# Pragmatic Fairness: Developing Policies with Outcome Disparity Control

## Abstract

We introduce a causal framework for designing optimal policies that satisfy fairness constraints. We take a pragmatic approach asking what we can do with an action space available to us and only with access to historical data. We propose two different fairness constraints: a moderation breaking constraint which aims at blocking moderation paths from the action and sensitive attribute to the outcome, and by that at reducing disparity in outcome levels as much as the provided action space permits; and an equal benefit constraint which aims at distributing gain from the new and maximized policy equally across sensitive attribute levels, and thus at keeping pre-existing preferential treatment in place or avoiding the introduction of new disparity. We introduce practical methods for implementing the constraints and illustrate their uses on experiments with semi-synthetic models.

## 1 Introduction

The fairness of decisions made by machine learning models involving underprivileged groups has seen increasing attention and scrutiny by the academic community and beyond. A growing body of literature has been looking at the unfavourable treatment that might arise from historical biases present in the data, data collection practices, or the limits of modelling choices and techniques. Within this field of study, the vast majority of works has considered the problem of designing *fair prediction systems*, i.e. systems whose outcomes satisfy certain properties with respect to membership in a sensitive group (Verma & Rubin, 2018; Barocas et al., 2019; Mehrabi et al., 2019; Wachter et al., 2020; Mitchell et al., 2021; Pessach & Shmueli, 2022). In contrast, relatively little attention has been given to the problem of designing *fair optimal policies* (Joseph et al., 2016; 2018; Gillen et al., 2019; Kusner et al., 2019; Nabi et al., 2019; Chohlas-Wood et al., 2021). In this case, the goal is to design a decision making system that specifies how to select *actions* that maximize a *downstream outcome* of interest subject to fairness constraints.

The consideration of an outcome downstream of the action allows a more flexible and powerful approach to fair decision making, as it enables to optimize future outcomes rather than matching historical ones, and also to enforce fairness constraints on the effects of the actions rather than on the actions themselves. For example, when deciding on offering college admissions, a fair prediction system would output admissions that match historical ones and satisfy certain properties with respect to membership in a sensitive group. In contrast, a fair optimal policy system would prescribe admissions such that downstream outcomes, e.g. academic or economic successes, are maximised and satisfy certain properties with respect to membership in a sensitive group. Enforcing fairness constraints on the effects of the actions in decision problems that correspond to allocations of resources or goods may reduce the risk of unfair delayed impact Dwork et al. (2012); Liu et al. (2018); D'Amour et al. (2020).

In this work, we propose a causal framework for designing fair optimal policies that is inspired by the public health literature (Jackson & VanderWeele, 2018; Jackson, 2018; 2020). We assume the causal graph in figure 1a, where action $A$ depends on sensitive attribute $S$ and covariates $X$; and where $S$ and $X$, which can be associated, potentially directly influence the outcome $Y$. We define the policy with the conditional distribution $p(A \mid S, X; \sigma_A)$ parameterized by $\sigma_A$–represented in the graph as a *regime indicator* (Dawid, 2007; Correa & Bareinboim, 2020). The aim is to learn a parametrization that maximizes $Y$ in expectation

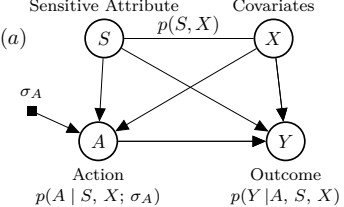 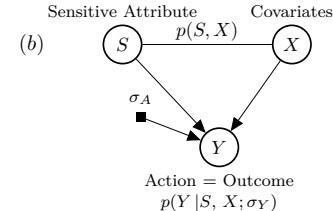

Figure 1: (a): Causal graph with associated distribution $p(A, S, X, Y; \sigma_A)$ = $p(Y|A, S, X)p(A|S, X; \sigma_A)p(S, X)$ describing our fair optimal policy setting. (b): A schema of the fair prediction system setting.

while controlling its dependence on $S$. This graph describes common real-world settings in which the action can only indirectly control for the dependence of $Y$ on $S$, and therefore to a level that depends on the available action space. In addition, it enables to learn the optimal policy from available historical data collected using a baseline policy $p(A \,|\, S, X; \sigma_A = \emptyset)$, overcoming issues such as, e.g., cost or ethical constraints in taking actions, which are common in real-world applications where fairness is of relevance. To gain more insights into the difference between our fair optimal policy framework and the fair prediction system framework, we offer a schema of the fair prediction setting with the casual graph in figure 1b, in which the action coincides with the outcome; and describe the goal as learning a parametrization such that $p(Y \,|\, S, X; \sigma_Y)$ matches the distribution of past outcomes $p(Y \,|\, S, X; \sigma_Y = \emptyset)$ while controlling for dependence of the outcome distribution on $S$.

We consider two constraints for controlling disparity in the downstream outcomes with respecct to $S$, which may be applicable for different use cases, depending on context and available actions: (i) a moderation breaking constraint, which aims at actively reducing disparity to the extent permitted by the action space; and (ii) an equal benefit constraint, which aims at equalizing the disparity of a new policy with that of the baseline policy in order to maintain the preferential treatment present in the baseline policy or to conservatively avoid introducing new disparity. We demonstrate the performance of our framework on two real-world based settings.

## 2 Outcome Disparity Controlled Policy

We assume the setting represented by the causal graph in figure 1a, where the sensitive attribute $S$ corresponds to characteristics of an individual–such as race, gender, disabilities, sexual or political orientation– which we wish to protect against some measure of unfairness. We focus on discrete $S$, but the proposed methods can be adapted to settings with continuous $S$.

This causal graph encodes the following statistical independencies assumptions: (a) $\sigma_A \perp\!\!\!\perp Y \,|\, \{A, S, X\}$; (b) $\sigma_A \perp\!\!\!\perp \{S, X\}$. These assumptions are not restrictive, as they are satisfied in any setting in which the partial ordering among nodes is given by $\{S \cup X, \sigma_A, A, Y\}$. Assumption (a) enables us to learn the optimal policy based on historical data collected using a *baseline policy* $p(A \,|\, S, X; \sigma_A = \emptyset)$ (observational-data regime), rather than by taking actions (interventional-data regime). Such a baseline policy represents the action allocation that was in place during the collection of the data pre-optimization. Assumption (b) allows us to compute the constraints with the proposed estimation methods below.

We denote with $Y_{\sigma_A}$ the *potential outcome* random variable, which represents the outcome resulting from taking actions according to $p(A \,|\, S, X; \sigma_A)$ and has distribution equal to $p(Y; \sigma_A) = \int_{a,s,x} p(a, s, x, Y; \sigma_A)$. In the reminder of the paper, we indicate the baseline policy and potential outcome with $p(A \,|\, S, X; \emptyset)$ and $Y_\emptyset$ respectively.

The goal of the decision maker is to learn a parametrization $\sigma_A$ that maximizes the expectation of the potential outcome, $\mathbb{E}[Y_{\sigma_A}]$, while also controlling the disparity in $Y_{\sigma_A}$ across $S$ in the following ways: (i) through a *moderation breaking* (ModBrk) constraint the aims at removing dependence of $\mathbb{E}[Y_{\sigma_A}]$ on $S$ as much as possible via what can be controlled, namely the allocation of $A$ as determined by $p(A|S, X; \sigma_A)$;

(ii) through an *equal benefit* (`EqB`) constraint that requires the distribution of $Y_{\sigma_A} - Y_\emptyset$ to be approximately equal across different values of $S$, ensuring that gain from the new policy is distributed equally.

## 2.1 Disparity Control via `ModBrk` Constraint

Without loss of generality[1], $\mu^Y(a, s, x) := \mathbb{E}[Y \mid a, s, x]$ can be written as

$$\mu^Y(a, s, x) = f(s, x) + g(a, s, x) + h(a, x), \tag{1}$$

which leads to the following decomposition of $\mu^Y_{\sigma_A}(s, x) := \mathbb{E}[Y_{\sigma_A} \mid s, x] = \int_a \mu^Y(a, s, x) p(a \mid s, x; \sigma_A)$

$$\mu^Y_{\sigma_A}(s, x) = f(s, x) + g_{\sigma_A}(s, x) + h_{\sigma_A}(x),$$

where $g_{\sigma_A}(s, x) := \mathbb{E}[g(A, s, x) \mid s, x; \sigma_A]$, and $h_{\sigma_A}(x) := \mathbb{E}[h(A, x) \mid s, x; \sigma_A]$. This decomposition contains (i) a component $f(s, x)$ that cannot be affected by $\sigma_A$, but which can contribute to disparity; (ii) a component $h_{\sigma_A}(x)$ that can be adjusted by $\sigma_A$ to increase expected outcomes, but which is not affected by $S$; and (iii) a component $g_{\sigma_A}(s, x)$ by which the choice of $\sigma_A$ can influence differences that are *moderated* by $S$. Considering how $\mu^Y_{\sigma_A}(s, x)$ varies with $s$ as a measure of disparity suggests the constrained objective

$$\arg\max_{\sigma_A} \mathbb{E}[Y_{\sigma_A}] \quad \text{s.t.} \quad (\mathbb{E}[g_{\sigma_A}(s, X) \mid S = s] - \mathbb{E}[g_{\sigma_A}(\bar{s}, X) \mid S = \bar{s}])^2 \leq \epsilon, \forall s, \bar{s}. \tag{2}$$

The slack $\epsilon$ is chosen based on domain requirements and the feasibility of the problem: central to the setup is that we work with a given space of policies, which is constrained by real-world phenomena and, in general, can only do so much to reduce disparity.

**Motivation & Suitability.** `ModBrk` aims at removing $S$'s influence on the outcome. As such, it is appropriate when disparity is illegitimate. This is analogous to the appropriateness of the demographic parity constraint in fair prediction systems.

However, `ModBrk` controls disparity via the separate actions and therefore to the extent permitted by the action space. Some insights into `ModBrk` can be gained by noticing that we are operating in the (estimated) true process that follows a particular $\sigma_A$. We marginalize the intermediate process between $(A, S, X)$ and the outcome $Y$, but implicitly assume that actions change mediating events, such as $M$ in figure 2a. The extent by which we can mitigate unfairness is a property of the real-world space of policies. For instance, we cannot interfere in the direct dependence between $S$ and $Y$ except for the moderating effect that $M$ might have under a new policy. This is in contrast to previous work, such as Nabi et al. (2019), which solves a planning problem in a "projection" of the real-world process where unfair information has been removed by some criteria.

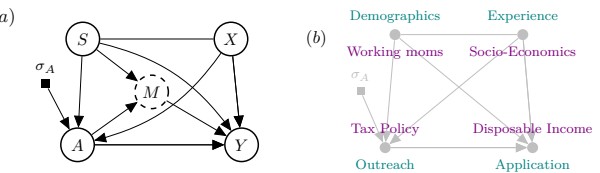

Figure 2: (a): Causal graph providing further intuition for the `ModBrk` constraint. (b): Example use cases for the suggested constrains. See full description in the text.

An example in which the `ModBrk` constraint would be suitable is that of a company looking to change its outreach campaign $A$ to maximize job applications $Y$ while mitigating current imbalances in their demographics $S$ and level of experience $X$ (figure 2b, green labels). The company cannot control factors such as cultural preference among applicants for industry sectors, but can induce modifications by focusing recruiting efforts in events organized by minority groups in relevant conferences and by choosing recruiting strategies that do not interact with group membership.

---

[1]This can be seen by setting $f(s, x) = h(a, x) = 0$.

## 2.2 Disparity Control via `EqB` Constraint

We formalize the requirement that the distribution of $Y_{\sigma_A} - Y_\emptyset$ must be approximately equal across different values of $S$ using the cumulative distribution function (cdf) $\mathcal{F}$, leading to the constrained objective

$$\arg\max_{\sigma_A} \mathbb{E}[Y_{\sigma_A}] \quad \text{s.t.} \quad \mathcal{F}(Y_{\sigma_A} - Y_\emptyset \mid S = s) = \mathcal{F}(Y_{\sigma_A} - Y_\emptyset \mid S = \bar{s}), \forall s, \bar{s}. \tag{3}$$

In general, the distribution of $Y_{\sigma_A} - Y_\emptyset$ is not identifiable, due to the fact that the decision maker has access only to realization of $Y$ under the baseline policy. This problem is similar to the "fundamental problem of causal inference" in the context of hard interventions, for which bounding approaches have been suggested Pearl (2000); Wu et al. (2019); Miratrix et al. (2018). Similarly, we propose matching upper bounds and lower bounds of the cdf.

**Motivation & Suitability.** `EqB` aims at not increasing disparity compared to the baseline policy. The `EqB` constraint would be appropriate in two scenarios. The first scenario is when we would like to keep the disparity present in the baseline policy when moving to the new one, since it includes legitimate disparity or desirable preferential treatment. This is a similar motivation to the equalized odds constraint in fair prediction systems. This scenario would arise, for example, when wishing to design an income taxation system to optimize disposable income or savings levels (figure 2b, purple labels). Disparities in $Y$ across gender or racial groups $S$ may exist, as well as possibly related individual characteristics $X$, like socio-economic status, employment type and housing situation. The decision maker should take into account and maintain desirable preferential treatments present in the baseline policy, for example, for working moms.

The second scenario is when we would like to not introduce greater disparity in a new policy, acknowledging the limitations of the action space. This scenario would arise, e.g., when wishing to perform a software update of a medical or well-being app that includes different designs, with users belonging to different age groups. The decision maker might want to ensure that the new policy does not change existing levels of difference in health or wellbeing across groups, which relates to the usage of the app. The app use, as well as the downstream outcome $Y$ itself, has pre-existing relations to age groups $S$ as well as to other associated user characteristics $X$, but the decision maker wants the update rollout to not exacerbate the already existing gap in outcome under the baseline policy. The decision maker may not want to forcefully reduce disparities in outcome in such a case – it is likely that younger age groups would be more responsive and happier with a software update, and have higher levels of the downstream outcome $Y$ overall; the choice of policy can only do so much to change that.

## 3 Relation to Prior Work

While more attention has been given to the problem of designing fair prediction systems, some works have considered the problem of designing fair optimal policies. Like us, Nabi et al. (2019) use observational data, but in order to learn a policy as if particular path-specific effects between $S$ and $A$ and $S$ and $Y$ were completely deactivated. They do not place any constraints on the distribution of $Y_{\sigma_A}$ given $S$ and $X$, require complex counterfactual computations, and aim to achieve a notion of fair policy which targets specific causal subpaths. Critically, they rely on a manipulation of $S$, and do not have an equivalent to our pragmatic view, asking what can be done with a set of available actions. Chohlas-Wood et al. (2021) consider a more general utility function than ours as the optimization objective but focuses on enforcing a notion similar to demographic parity on the choice of the policies, rather than considering a fairness notion on the outcomes. The most similar work to ours is the one from Kusner et al. (2019), still in the observational data regime. Crucially, this work is different in its motivation: it aims to extend the work in Kusner et al. (2017) to the policy setting, and thus relies on manipulation of $S$. Further, it consists mostly of budget treatment allocations and interference problems. It does not have our pragmatic view, and does not consider parameterized policy spaces that take into account a covariate vector $X$. Due to the difference in their disparity definitions, these methods would not be directly comparable to ours. There is also a growing literature on fair policy optimization in online settings, but those deal with sequential decisions and interventional data, which are fundamentally different from ours (see Appendix A).

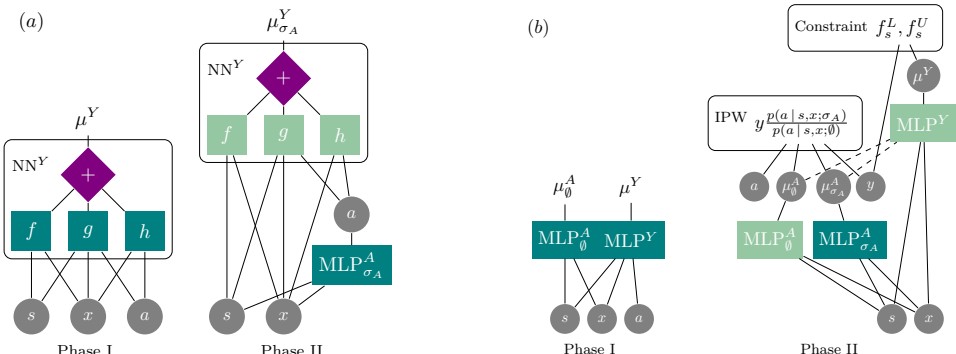

Figure 3: (a) Training phases for `ModBrk`. (b) Training phases for `EqB`. Gray circles: inputs. Teal blocks: parameter layers. Light green blocks: fixed parameters. Purple diamonds: additive gates. Dashed edge: alternating inputs.

# 4 Method

In this section, we describe how the `ModBrk` and `EqB` constrained objectives (2) and (3) can be estimated from an observational dataset $\mathcal{D} = \{a^i, s^i, x^i, y^i\}_{i=1}^N$, $(a^i, s^i, x^i, y^i) \sim p(A, S, X, Y; \emptyset)$, and introduce a method for optimizing $\sigma_A$ using neural networks for each constraint. In both cases, we enforce the constraints via an augmented Lagrangian approach, casting them as inequality constraints controlled by a slack value $\epsilon$ (see Appendix B). Different choices of $\epsilon$ lead to different trade-offs between utility and constraints, similarly to varying Lagrange multiplier values.

## 4.1 `ModBrk` Constraint

For the `ModBrk` constrained objective (2), we learn a deterministic policy using an MLP neural network $\text{MLP}_{\sigma_A}^A$, i.e. $p(A \mid s, x; \sigma_A) = \delta_{A=\text{MLP}_{\sigma_A}^A(s,x)}$, where $\delta$ denotes the delta function and $\text{MLP}_{\sigma_A}^A(s, x)$ the output of $\text{MLP}_{\sigma_A}^A$ given input $s, x$. This gives $\mu_{\sigma_A}^Y(s, x) = \mathbb{E}[Y_{\sigma_A}|s, x] = \int_a \mu^Y(a, s, x) p(a \mid s, x; \sigma_A) = \mu^Y(\text{MLP}_{\sigma_A}^A(s, x), s, x)$.

We model $\mu^Y(a, s, x) = f(s, x) + g(a, s, x) + h(a, x)$ using a structured neural network $\text{NN}^Y$ that separates into the three components $f, g, h$ reflecting the decomposition of $\mu^Y(a, s, x)$.

We learn the parameters of $\text{MLP}_{\sigma_A}^A$ and $\text{NN}^Y$ in two phases, as outlined in figure 3a. In Phase I, we estimate the parameters of $\text{NN}^Y$ from $\mathcal{D}$. In Phase II, we learn the parameters of $\text{MLP}_{\sigma_A}^A$ by optimizing objective (2) with $\mathbb{E}[Y_{\sigma_A}] \approx \frac{1}{N} \sum_{i=1}^N \mathbb{E}[Y_{\sigma_A} \mid s^i, x^i] = \frac{1}{N} \sum_{i=1}^N \mu^Y(\text{MLP}_{\sigma_A}^A(s^i, x^i), s^i, x^i)$, using the $\text{NN}^Y$ trained in Phase I. If all variables are discrete, after estimating $\mu^Y(a, s, x)$ and $p(s, x)$, computing (2) can be cast as an LP (see Appendix C). Consistency is then a standard result which follows immediately. Notice that this formulation can also be extended to continuous $S$ by defining the constraint via partial derivative, $\mathbb{E}\left[ \left| \frac{\partial g_{\sigma_A}(s, X)}{\partial s} \right| \Big| S = s \right] \leq \epsilon$.

**Action clipping.** We suggest ensuring overlap of $p(a|s, x; \sigma_A)$ with $p(a|s, x; \emptyset)$ by adaptively constraining the output of $\text{MLP}_{\sigma_A}^A$ to be within an interval that resembles the one observed under the baseline policy. We consider two options: (a) matching the minimal and maximal value $a$ seen for each $s, x$ combination (binning continuous elements) and (b): extend that interval according to the difference between the minimal and maximal $a$ value seen with each $s, x$. In the experiments, we opted for constraining the output of $\text{MLP}_{\sigma_A}^A$ to be in the interval $[\min_{A_{S,X}} - \eta \text{gap}_{A_{S,X}}, \max_{A_{S,X}} + \eta \text{gap}_{A_{S,X}}]$, where $\text{gap}_{A_{S,X}} = \max_{A_{S,X}} - \min_{A_{S,X}}$. We enforced this interval with a shifted Sigmoid function in the last layer of $\text{MLP}_{\sigma_A}^A$. We set $\eta = 1$ to allow some extrapolation and increase in utility.

## 4.2 `EqB` **Constraint**

For the `EqB` constrained objective (3), we require knowledge of the baseline policy as well as creating upper and lower bounds on the cdf $\mathcal{F}(Y_{\sigma_A} - Y_\emptyset \,|\, S = s), \forall s$. $Y_{\sigma_A} - Y_\emptyset$ is a counterfactual quantity, which is not generally identifiable. In fact, we do not have access to the joint distribution of $Y_{\sigma_A}$ and $Y_\emptyset$ as we never observed both variables simultaneously for the same individual. However, we show next that if one is willing to make a parametric assumption, then $Y_{\sigma_A} - Y_\emptyset$ is partially identifiable and amenable to bounding. We assume that the baseline policy is Gaussian and only consider Gaussian policies, with equal homogeneous variance, i.e. $p(A|s,x;\emptyset) = \mathcal{N}(\mu_\emptyset^A(s,x), V^A)$, $p(A|s,x;\sigma_A) = \mathcal{N}(\mu_{\sigma_A}^A(s,x), V^A)$. In addition, we assume that $Y_{\sigma_A}$ and $Y_\emptyset$ are jointly Gaussian conditioned on $S, X$, with marginal means $\mu_{\sigma_A}^Y(s,x)$, $\mu_\emptyset^Y(s,x)$ and equal homogenous marginal variance $V^Y$.

The assumption on $Y_{\sigma_A}$ and $Y_\emptyset$ enables us to bound the population cdf by maximizing/minimizing it with respect to the unknown correlation coefficient $\rho(s,x) \in [-1,1]$, where $\rho(s,x) := \frac{Cov(Y_{\sigma_A}, Y_\emptyset \mid s,x)}{V^Y}$. Let $\Phi$ denote the cdf of the standard Gaussian. We can write

$$\mathbb{P}(Y_{\sigma_A} - Y_\emptyset \leq z \,|\, x,s) = \Phi\left(\frac{z - \mu_{\sigma_A}^Y(s,x) + \mu_\emptyset^Y(s,x)}{\sqrt{2V^Y(1 - \rho(s,x))}}\right).$$

Defining $\mu(s,x) := \mu_{\sigma_A}^Y(s,x) - \mu_\emptyset^Y(s,x)$ and $f_{s,x}^z(\rho) := \Phi\left(\frac{z - \mu(s,x)}{\sqrt{2V^Y(1-\rho)}}\right)$, we can show that, for any $s, x$, the following bounds are the tightest: (i) $f_{s,x}^z(-1) \leq \mathbb{P}(Y_{\sigma_A} - Y_\emptyset \leq z \,|\, x,s) \leq f_{s,x}^z(1)$, for $z - \mu(s,x) > 0$; (ii) $f_{s,x}^z(1) \leq \mathbb{P}(Y_{\sigma_A} - Y_\emptyset \leq z \,|\, x,s) \leq f_{s,x}^z(-1)$, for $z - \mu(s,x) < 0$.

Using $N_s$ to indicate the number of elements in $\mathcal{D}$ with $s^i = s$, we obtain global lower and upper bounds estimates for $\mathbb{P}(Y_{\sigma_A} - Y_\emptyset \leq z \,|\, s)$ as $F_s^L(z) = \frac{1}{N_s} \sum_{i:s^i=s} f_{s,x^i}^z(-\text{sign}(z - \mu(s,x^i)))$ and $F_s^U(z) = \frac{1}{N_s} \sum_{i:s^i=s} f_{s,x^i}^z(\text{sign}(z - \mu(s,x^i)))$. We operationalize the constraint by minimizing the mean squared error (MSE) of the bounds differences, i.e. our final objective is

$$\arg\max_{\sigma_A} \mathbb{E}[Y_{\sigma_A}] \quad \text{s.t.} \quad \sum_{z \in S_z} \left(||F_s^L(z) - F_{\bar{s}}^L(z)||_2^2 + ||F_s^U(z) - F_{\bar{s}}^U(z)||_2^2\right) \leq \epsilon, \tag{4}$$

$\forall s, \bar{s}$, where $S_z$ is a grid of values.

We propose to estimate $\mathbb{E}[Y_{\sigma_A}]$ with the inverse probability weighting (IPW) estimator. This is an alternative estimation approach to the one used for `ModBrk`, to demonstrate the modularity of our approach and constraints.

$$\mathbb{E}[Y_{\sigma_A}] = \int_{a,s,x,y} y \frac{p(a \,|\, s,x;\sigma_A)}{p(a \,|\, s,x;\emptyset)} p(a,s,x,y;\emptyset) \approx \frac{1}{N} \sum_{i=1}^N y^i \frac{p(a^i \,|\, s^i, x^i; \sigma_A)}{p(a^i \,|\, s^i, x^i; \emptyset)}.$$

We model $\mu_\emptyset^A(s,x)$, $\mu_{\sigma_A}^A(s,x)$ and $\mu^Y(a,s,x) = \mathbb{E}[Y \,|\, a,s,x]$ using MLP neural networks $\text{MLP}_\emptyset^A$, $\text{MLP}_{\sigma_A}^A$, and $\text{MLP}^Y$ respectively. We learn the parameters of these networks in two phases as outlined in figure 3b. In Phase I, we learn the parameters of $\text{MLP}_\emptyset^A$ and $\text{MLP}^Y$ from $\mathcal{D}$ with MSE losses between predicted and observed actions and outcome. We obtain a MAP estimate of $\mu_\emptyset^Y(s,x) = \int_a \mu^Y(a,s,x)p(a|s,x,\emptyset)$ as $\hat{\mu}_\emptyset^Y(s,x) = \int_a \mu^Y(a,s,x)\delta_{A=\mu_\emptyset^A(s,x)}$. We estimate $V^A$ and $V^Y$ through averaging the MSE of target and predicted mean output from $\text{MLP}^A$ and $\text{MLP}^Y$ respectively [2]. In Phase II, we learn the parameters of $\text{MLP}_{\sigma_A}^A$ that maximize objective (4) using the $\text{MLP}_\emptyset^A$ and $\text{MLP}^Y$ trained in Phase I.

**Misspecification and parametric restrictiveness.** Notice that, as the Gaussianity assumption is placed on the conditional joint distribution of $(Y_{\sigma_A}, Y_\emptyset)$ given $S$ and $X$ rather than on the marginal distribution. It is thus reasonable to invoke the central limit theorem and assume that, given enough samples, the residuals of $Y_\emptyset|s,x$ and $Y_{\sigma_A}|s,x$ are normally distributed, since one can view the residuals as aggregating various minor factors that explain the remaining variability after taking into account a strong signal. Also notice

---

[2] We assume $V^Y$ to be homoskedastic.

that our proposed approach can be easily extended to heteroskedastic Gaussian regression models which are very flexible and can accommodate many real-world datasets. Finally, a nonparametric approach for the constraint computation would be possible e.g. via Fréchet bound (see Appendix D), but that may come at a computational cost. To illustrate the effects of misspecification on our proposed assumption, we present below a formal result showing that our constraint estimation error is bounded from above by how the true joint density deviates from the Gaussian density (see Appendix D for a proof and discussion).

**Proposition 1** *For fixed $s, x$, let $Q = \mathbb{P}(Y_{\sigma_A} - Y_\emptyset \leq z \,|\, s, x)$ and our estimator $\hat{Q} = \Phi\left(\frac{z - \mu_{\sigma_A}^Y(s,x) + \mu_\emptyset^Y(s,x)}{\sqrt{2V^Y(1-\rho(s,x))}}\right)$. Let $f_{s,x}$ denote the joint density function of $Y_{\sigma_A}, Y_\emptyset \,|\, s, x$ and $\phi_{s,x}$ the joint Gaussian density function with mean $[\mu_{\sigma_A}^Y(s,x), \mu_\emptyset^Y(s,x)]$ and shared variance $V$ with correlation $\rho(s,x)$. Then $\left|Q - \hat{Q}\right| \leq \|f_{s,x} - \phi_{s,x}\|_1$, where $\|\cdot\|_1$ denotes the $L_1$ norm.*

## 5 Experiments

We evaluated the `ModBrk` and `EqB` constraint methods[3] on the New York City Public School District (NYC-Schools) dataset compiled in Kusner et al. (2019), which we augmented to include actions; and further tested `ModBrk` on the Infant Health and Development Program (IHDP) dataset, specifically the real-data example examining dosage effects described in Section 6.2 of Hill (2011) (we could not use this dataset for `EqB` due to its reliance on parametric assumptions). Below, we briefly discuss the datasets and defer further details to Appendix E.

**Action-augmented NYCSchools dataset.** We adopted the same sensitive attribute and covariates as Kusner et al. (2019), and augmented the dataset with generated actions and outcomes. We created continuous actions corresponding to funding level decisions as $A = (w_{SX}^T SX)^2 + \max(0, w_X^T X) + \mathcal{N}(0.5, 0.4)$, where $E$ is the original percent of students taking the SAT/ACT exams (pre-college entry) in the dataset. The proposed form of $Y$ is discussed in Appendix E.

**IHDP dataset.** The IHDP dataset describes a program that targeted low-birth-weight, premature infants, providing them with intensive high-quality child care and home visits from a trained provider. As continuous action $A$ we used the self-selected number of participation days in the program. The outcome $Y$ corresponds to child score attainment in cognitive tests at age three. We considered mother's race (white vs. non-white) as sensitive attribute $S$. We reinterpreted the original setting as a resource allocation problems as follows: rather than as a self-decision, we view the number of days in treatment as external, e.g., by assigning different individuals to different lengths of participation. In this case, the `ModBrk` constraint goal is to break the moderation of the allocation of days in program by the group membership, such that the resource allocation policy is not responsible for the difference in scores attained by different groups.

### 5.1 Results

We evaluated `ModBrk` and `EqB` by comparing these methods with: (1) optimizing the policy with no disparity constraint (Unconstrained), (2) optimizing the policy without using $S$ (Drop $S$), and (3) using the baseline policy ($\sigma_A = \emptyset$). As discussed in Section 2, previous work on fair policy or action allocation differs from our setting and objective, and therefore cannot be meaningfully compared. For `ModBrk`, we also compare against different levels of constant actions (Const$_A$). We do not include it for `EqB` since the IPW estimator is not suitable for $\delta$-distributions. The range of $\epsilon$ values in the plots was determined such that the minimal $\epsilon$ slack values correspond to the smallest constraint value achievable with our optimization strategy (this is the lowest possible constraint value achieved when setting $\epsilon = 0$); the maximal $\epsilon$ slack value is the one that would closely match the constraint violation under unconstrained optimization. In practice, we propose to explore the frontier resulting from choices of slack values and select the most acceptable trade-off for the user.

---

[3]The code reproducing our results is available at `https://github.com/limorigu/PragmaticFairness`

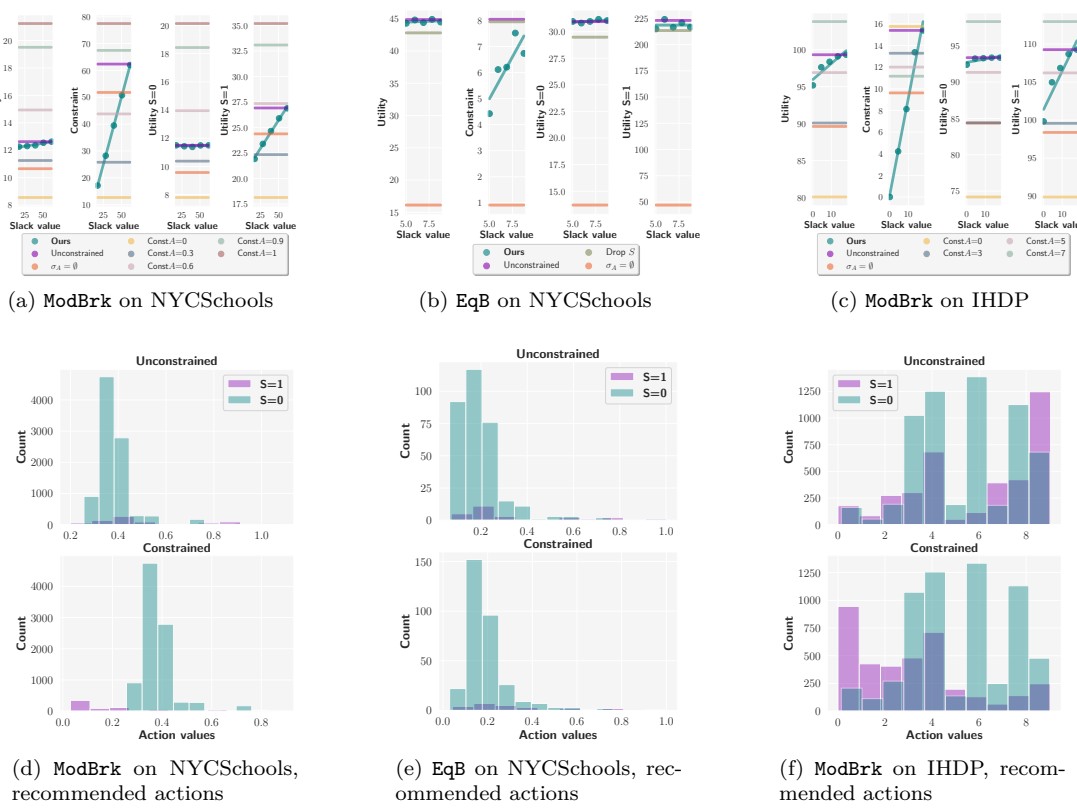

(a) `ModBrk` on NYCSchools  (b) `EqB` on NYCSchools  (c) `ModBrk` on IHDP

(d) `ModBrk` on NYCSchools, recommended actions  (e) `EqB` on NYCSchools, recommended actions  (f) `ModBrk` on IHDP, recommended actions

Figure 4: (Top) influence of the slack value on obtained values of the objective function and fairness constraint violations. (Bottom) recommended actions by group for unconstrained optimization and for constrained optimization, where the slack on fairness violation set to 0 or the highest level shown in the top plot, respectively.

**ModBrk Constraint Method.** The results for `ModBrk` on the NYCSchools and IHDP datasets are presented in Fig. 4a, 4d and in Figs. 4c, 4f respectively[4]. For both datasets, as we increase the tolerance on fairness violations, in the form of higher slack value $\epsilon$ in (2), we see a major increase in constraint value, while allowing some increase in utility $\mathbb{E}[Y_{\sigma_A}]$. Observing the utility values broken down by group membership ($\mathbb{E}[Y_{\sigma_A} \,|\, S = 1]$ vs. $\mathbb{E}[Y_{\sigma_A} \,|\, S = 0]$), we see most of this increase of utility is driven by the privileged group, $S = 1$ (a majority-white student body in the NYCSchools dataset, and white mothers for the IHDP dataset). This is to be expected given that we are trying to minimize interactions involving $S = 1$ and $A$, i.e. where the membership in the privileged group inflates utility values, via higher $g$ values. These higher $g$ values for the $S = 1$ group also translate into higher utility values for $S = 1$ according the baseline policy, as can be seen from the orange line, indicating baseline policy actions in the corresponding figures. Notice that the choice of an appropriate slack $\epsilon$ here is a matter of trade-off based on user preferences of utility vs. constraint value, and will depend on the specific dataset. We can gain deeper insight into the working of our approach by inspecting the histogram of recommended actions at the end of the policy model training. For both datasets, we see that applying the constraint with a small $\epsilon$ value results in mapping more $S = 1$ members to lower action values compared to the $S = 0$ group, indicating that to enforce the constraint more tightly means allocating lower value actions to the privileged group, while giving out as many high actions to $S = 0$ as possible under the action clipping setting described previously. Notice that in both settings we succeed in increasing the utility compared to the baseline policy. Recall that we employ action clipping to ensure some overlap with the baseline actions for estimation purposes. This means that we are bound to

---

[4]Dropping $S$ had no influence here on utility and constraint values, thus we removed this baseline to avoid confusion due to overlap and subsequent mismatch of colors between figure and legend.

some extent to the interval of actions seen for each $s, x$ in the baseline policy (this trade-off can be explored via choice of $\eta$ value, see Section 4.1). One could also increase $\eta$ to achieve higher utility at the cost of increased total action allocation (i.e. higher "budget") and sacrifice in coverage. This is also why we cannot achieve utility values that are as high as the highest $\text{Const}_A$ in the figures; we ask how best to distribute actions within the effective "budget".

**EqB Constraint Method.** We present results for `EqB` on the NYCSchools dataset in Figs. 4b and 4e. As we increase the tolerance on fairness violations - in the form of higher slack value $\epsilon$ in (4) - we observe an increase in constraint value with almost no change in overall utility $\mathbb{E}[Y_{\sigma_A}]$. Observing the utility values broken down by group membership ($\mathbb{E}[Y_{\sigma_A} \mid S = 1]$ vs. $\mathbb{E}[Y_{\sigma_A} \mid S = 0]$), we see also no significant change in utility coming from either group. This result indicates that our method learns a policy that assigns actions ensuring equal benefit without sacrificing the utility of either group. Note that we also see an unusually high estimate of utility for $S = 1$. This could be explained by the IPW estimator not extrapolating well to unseen data, as the $S = 1$ group only consists of $7\%$ of an already small dataset. On the right hand side of Fig. 4b we observe similar recommended action histograms for the unconstrained (top right) and constrained (bottom right) cases. This shows that our method decreases disparity with a similar "budget". This is possible because we are not attempting to reduce disparity present under the baseline policy. The similarity in action histograms is partially due to the IPW objective which aligns recommended actions with observed ones to achieve higher weighting. To validate our approach, we compute the ground truth constraint value through counterfactual realization of the policy model's predicted action mean in the data generating process and compute distributional difference of $Y_{\sigma_A} - Y_\emptyset \mid S$ for the different sensitive attribute groups. Notice also that our method succeeds in increasing the utility compared to the baseline policy. Although the baseline policy has the lowest constraint, this is expected as we are comparing distributions of $Y_{\sigma_A} - Y_\emptyset \mid S$, where $\sigma_A = \emptyset$. Dropping $S$ has a slight increase in constraint and a decrease in utility compared to unconstrained setting, as our policy model performs better through taking into account $S$ to break the indirect association between $S$ and $Y$.

## 6    Conclusion

We introduced a causal framework to learn fair policies given access to observational data and an action space. Taking a pragmatic view, we asked what is the best utility that can be achieved with the provided action space, while controlling two notions of disparity: one focusing on mitigating a possible moderation effect involving group membership and the policy, and the other focusing on ensuring equal benefit with respect to a baseline policy. We see this work as a first conceptual contribution in defining pragmatic fair impact policies, and envisage various possible future directions, including extending the proposed methods beyond binary sensitive attributes, to a multi-stage policy setting, to handle unmeasured confounding and to online optimization.

**Limitations discussion.** *Conceptual.* Our pragmatic approach relies on data that we have about mechanisms *in this world*, including possible inequities our actions in question cannot affect. We also do so within a stationary framework that currently does not account for feedback, although we see it as an important next step. This comes with caveats. First, whether or not it is desirable to control for disparities of outcomes across levels of the sensitive attribute is problem-dependent. For instance, if the action space consists of solely two options, to give a medical treatment or not, it may be unclear why we should take into consideration group differences in recovery: other things being equal, we just want to maximize the number of lives saved. In contrast, if $Y$ is a relative measure, e.g., of wealth, pure wealth maximization may be judged to be harmful if disparities among groups in $S$ are exacerbated. In this case, we may settle for a scenario with less aggregated wealth if disparities are controlled. Such value judgements are *not* to be decided algorithmically. Our goal is to provide a formalization of disparity control *if* it is judged to be desirable, and to provide an estimate of *whether* different levels of control can be achieved under an acceptable loss of total expected outcome, *given* an action space that is, again, a property of the real-world. We simply provide a framework to examine what is possible. However, if our data reflects biased mechanisms existing in the world that is outside of reach for our actions, we will not be able to change those, and they will be reflected in estimates of what is possible. This is in contrast to methods that aim to model alternative fair worlds.

*Technical.* The parametric assumption we make for the operationalization of the `EqB` constraint is one that could be avoided and extended for greater generalization (e.g., via Fréchet bounds, see Appendix D.1). However, in this work we opted for a parametric assumption to focus on a general concpetual introduction. For the `ModBrk` constraint we propose a simple MLP estimation of the decomposition in Eq. 1. One could envision a more elaborate formulation that will avoid a possible failure mode, where $g$ simply subsumes $f$ and $h$. We did not observe empirically that this happens in practice. However, one could avoid such possible issue by including additional regularization or residual connections between the components of the MLP.

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
