# OpenReview forum: "Pragmatic Fairness: Developing Policies with Outcome Disparity Control"
_TMLR — Rejected by TMLR_

### Review · Reviewer_YCuy · 2023-03-07

**Summary Of Contributions:**

This paper studies the question of how to design fair decision policies in settings where there is a downstream outcome that the policy designer is interested in optimizing. Two new fairness definitions are proposed: moderation breaking (ModBrk) and equal benefit (EqB). The ModBrk constraint is intended to remove the influence of the sensitive attribute on the downstream outcome while maximizing that outcome, while the EqB constraint is intended to ensure that, under a new policy regime, disparities in the outcome of interest are not worse than under the status quo policy regime. In addition to proposing the new definitions, the paper proposes distinct methods for enforcing ModBrk and EqB in real-world algorithmic design problems. The application of these methods is demonstrated on two real-world examples.

**Audience:**

Yes

**Broader Impact Concerns:**

Because of its underspecification, I am concerned that the authors do not adequately consider the consequences of a practitioner deploying a ModBrk-constrained decision system in the real world. This is somewhat mitigated by the fact that the authors suggest that practitioners consider a range of possible policies, including the optimal unconstrained policy.

**Claims And Evidence:**

No

**Requested Changes:**

**Major Changes:**
In order of relative priority, I would ask the authors to improve the manuscript in the following ways:
* The underspecification of ModBrk is, in my view, a critical issue for the current submission. In order for ModBrk to be a meaningful fairness criterion, it needs to be based on a decomposition of \\(\\mathbb {E}[Y \\mid  a, s, x]\\) that is (1) unique, (2) identifiable, and (3) meaningful from a policy / fairness perspective, as detailed above.
* The authors should address the following issues in their experimental design and reporting of results:
    * Can the authors clarify how the results of the IHDP experiment should be interpreted in light of the fact that, in the original data, the number of days participated was chosen by participants after treatment, rather than directly assigned? In particular, since the estimates don't appear to correspond to a meaningful estimand, in what sense is the experiment successful?
    * In the middle panels of Plots (a), (b), and (c), what is indicated by "constraint"? The slack value, \\(\\epsilon\\), or some other quantity? In addition, in the case of the ModBrk experiments, the slack values are difficult to interpret without units. In particular, it is hard to understand whether the slack values represent a large or small amount of slack.
    * In Plot (b), how is it possible that the constraint fell when the slack value was relaxed? The set of policies satisfying the relaxed constraint is a superset of the set of policies satisfying the stricter constraint, so the maximum utility can only increase.
    *  Because the number of individuals with S = 0 vs. S = 1 is so different in the NYCSchools dataset, Plots (d) and (e) are very difficult to read. Can you facet and normalize the distributions so that the policy can be discerned for the S = 1 group?
* Given the similarities between the approach developed in this manuscript and existing approaches in the literature, it is important for the authors to more carefully (1) engage with the substantial existing literature on this topic, and (2) delineate differences between their approach and existing approaches. In the case of multi objective approaches to fairness, this is critical; in the case of offline policy evaluation, this is also important, although less critically so. Here is a (non-exhaustive) selection of articles which take a similar approach, and in comparison to which the authors should clarify their contribution:
    * Ogryczak, W., Wierzbicki, A., and Milewski, M. (2008). A multi-criteria approach to fair and efficient bandwidth allocation. Omega, 36(3):451 – 463.
    * Bertsimas, D., Farias, V., and Trichakis, N. (2012). On the efficiency-fairness trade-off. Management Science, 58:2234—-2250.
    * Cai, W., Gaebler, J., Garg, N., & Goel, S. (2020, February). Fair allocation through selective information acquisition. In Proceedings of the AAAI/ACM Conference on AI, Ethics, and Society (pp. 22-28).
    * Eisenhandler and Tzur (2019) The humanitarian pickup and distribution problem. Operations Research, 67:10–32.
    * Mostajabdaveh, M., Gutjahr, W. J., and Sibel Salman, F. (2019). Inequity-averse shelter location for disaster preparedness. IISE Transactions, 51(8):809–829.
    * Nilforoshan, H., Gaebler, J. D., Shroff, R., & Goel, S. (2022, June). Causal conceptions of fairness and their consequences. In International Conference on Machine Learning (pp. 16848-16887). PMLR.
* There appear to be substantial obstacles in identifying and measuring many critical quantities considered in this paper. The authors should, at a minimum: (1) clarify which elements of the DAG are observed, and which are unobserved; and (2) comment on the strength of the causal and statistical assumptions discussed above.

**Minor Changes:**
* On p. 1, why is the form of the decision policy assumed to be parametric? In general, this assumption doesn't appear to be used outside of Section 4.2.
* On p. 5, why is the continuous analogue of ModBrk given as a bound on the derivative, when the most obvious generalization is simply a uniform (i.e., \\(L^\\infty\\)) bound on the difference? In particular, when the bound only holds on the derivative, even though locally disparities must be small, globally large disparities can exist between different values of S.
* Why is Eq. (4), on p. 6, written in terms of \\(\\ell_2\\) norms, when the outcome \\(Y\\) is scalar?

**Possible Typos:**
* P1: "as it enables to optimize future outcomes" -> Enables one to optimize?
* P1: "We define the policy with the conditional distribution p(A | S, X; σA) parameterized by σA–represented in the graph as a regime indicator" -> \\(\\sigma_A\\) *and* represented?
* P2: Please clarify what the notation \\(\\sigma_A = \\emptyset\\) means.
* P4: "The most similar work to ours is the one from Kusner et al. (2019)" -> The most similar work to ours is Kusner et al. (2019)?
* P7: It's not really correct to refer to \\(\\hat Q\\) as an estimator, since it doesn't involve any random quantities and isn't estimated from data.

**Appendices:**
I was unable to locate appendices C through E.

**Strengths And Weaknesses:**

The major strength of this submission is the authors suggestion that practitioners treat constraints as part of a multi-objective function to be optimized in tandem with other objectives (i.e., the outcome \\(Y\\) here), rather than as constraints per se. In particular, the advice to "explore the frontier resulting from choices of slack values and select the most acceptable trade-off for the user" seems important. However, the current draft of the submission suffers from a number of important weaknesses detailed below.

**Underspecification of ModBrk.** The moderation breaking fairness constraint proposed in this submission depends on a decomposition of \\(\\mathbb {E}[Y \\mid  a, s, x]\\) into three terms: \\[ \\mathbb {E}[Y \\mid a, s, x] = f(s,x) + g(a,s,x) + h(a,x). \\] The fairness constraint consists in bounding disparities in the contribution of \\(g(a,s,x)\\) to the expectation of \\(Y_{\\sigma_A}\\)---i.e., the counterfactual outcome of interest that would result if the policy \\(\\sigma_a\\) were adopted. However, because this decomposition is neither unique nor identifiable, the constraint itself is vacuous. For any given policy \\(\\sigma_A\\), one could take \\(f(s,x) = \\mathbb E [Y_{\\sigma_A} \mid s, x]\\), \\(g(a,s,x) = \\mathbb {E}[Y \\mid  a, s, x] - f(s,x)\\) and \\(h(a,x) = 0\\), and the policy \\(\\sigma_A\\) would satisfy the fairness constraint. To make this constraint non-vacuous, one would need to determine a method of fixing the decomposition; to make the constraint meaningful, one would need to ensure that the components of the decomposition were actually meaningful, in the sense that disparities in \\(\\mathbb{E}[g(A_{\\sigma_A}, s, x) \\mid s, x]\\) are something that the policymaker or algorithm designer cared about. This shortcoming is acknowledged in the conclusion, but the suggestion that additional regularization—which could reduce the degree of underspecification in the decomposition—is not adequate because it is not clear why the resulting decomposition would be meaningful from a policy perspective.

**Alternative Approaches.** The submission formulates the problem of fair decision policies as a constrained optimization problem, where the constraint is a disparity of interest to the policy designer. As noted above, the authors suggest considering the tradeoffs between the fairness constraint and the objective by varying the slack variable and considering the resulting effect on the expected utility. A related approach—which is essentially implicit in the constrained Lagrangian reformulation of the problem in Section 4—is to incorporate the constraint directly into the objective function as a penalty with some weight, and to consider various solutions as the weight is varied. From this perspective, the approach developed in this submission is very similar to a broader literature. I have included examples of additional literature adopting this perspective below. It is important to clarify how these methods relate to these existing approaches. The problem of learning an optimal policy from observational data, which is a second central concern in this work, has also been extensively considered in the reinforcement learning literature. It is unclear to me how the methods the authors propose relate to existing methods for offline policy evaluation developed in that literature.

**Experiments.** There appear to be a number of issues with both the analysis and the presentation of the experiments in Section 5. In particular, the overall presentation of results is somewhat difficult to follow, and some of the results differ from what one would expect theoretically in surprising ways. Specific issues are enumerated below. More broadly, the fact that the number of days which mothers chose to spend in the intensive care program in the IHDP data is treated as assigned, rather than chosen by the participants themselves, makes the results fairly difficult to interpret. Because of this reinterpretation, it is unclear what the experiment demonstrates, since the estimates don't correspond to an actual quantity of interest.

**Strong Statistical and Causal Inference Assumptions.** It is not entirely clear which of the variables in the DAG model in Fig. 1 are intended to be observed, rather than latent. The subsequent analysis seems to assume that all of the variables are observed, i.e., that there is no unobserved confounding. This is a strong assumption that should be addressed more directly. In addition, some of the quantities in the EqB constraint, are not identifiable. The submission suggests using partial identification strategies; however, in practice, such bounds tend to be quite loose. The authors develop bounds based on joint normality in Section 4.2; this, too however, is a fairly strong assumption, and seems unlikely to hold even approximately in practice. (The assumption that the baseline policy is Gaussian is especially strange in light of the fact that many such decisions are likely to be made on the basis of thresholds or other deterministic rules.) In general, because of the strong statistical and causal inference assumptions, it is unclear whether the suggested criteria could actually be convincingly put into practice.

---

### Review · Reviewer_Z84d · 2023-03-16

**Summary Of Contributions:**

Proposes a framework for studying the causal effect of policies on different groups under new fairness constraints. The first, moderation breaking (ModBrk), requires that group differences in received positive outcomes *attributable to the policy* are reduced as much as possible. The second, equal benefit (EqB), requires that each groups's distribution of improvements over the status quo policy is approximately equal.

The "pragmatic" part of the proposed approach is the introduction of two important, under-discussed considerations:

- That the appropriate subject of fairness analysis should be the *effects* of actions, not the actions themselves. Many previous papers in this space proposed simplistic constraints on actions (e.g. "applications in each group should be issued loans at equal rates") without considering whether the effect of such a policy would be beneficial for the groups in question.
- That interventions need to operate in the real world. This means that the effects of the available actions could systematically vary by group, and there may not exist any policy that achieves parity in the outcomes we care about. Previous work often assumed that groups are actually the same (in the value they get from actions), or that treating them as if they were the same is the best way to produce good outcomes.



**Audience:**

Yes

**Broader Impact Concerns:**

There is work that needs to be done to defend the proposed fairness criteria on normative/ethical grounds, but I don't think a broader impact statement is necessary for this mostly-theoretical paper.

**Claims And Evidence:**

No

**Requested Changes:**

### Critical to securing my recommendation for acceptance:

Most importantly, this paper needs a compelling worked example that readers can follow throughout. The example should highlight the value of the pragmatic considerations introduced by this paper, and any constraints introduced should be clearly defensible in this setting.

Second, the constraints need to be explained and justified more clearly. ModBrk is introduced as "appropriate when disparity is illegitimate" and analogized to demographic parity. But ModBrk actually allows disparities to persist, even when they could be reduced further by the policy (since ModBrk minimizes disparities *due to the policy*, not overall).

Furthermore, I don’t think ModBrk is really analogous to demographic parity, because the outcome being equalized here is something like “Net benefit delivered by the policy”, not “fraction receiving the decision Y=1”. The vast majority of demographic parity papers do not have a notion of benefit or wellbeing, they simply seek independence between group and decision. It seems to me that this is the crucial differentiator of this paper: it understands that not all disparities in wellbeing can be mitigated with a given decision, and that not all decisions are equally valuable for different people. Applied to lending, demographic parity means equal loan acceptance rates for applicants from all groups. ModBrk almost certainly wouldn’t produce this solution, because not all applicants benefit from being issued loans (since default is very costly for borrowers) and the distribution of those who could benefit might not be the same for all groups.

EqB seems less defensible than ModBrk, as it essentially formalizes status quo bias with respect to disparities. For example, consider a setting where the optimal policy (from the decision-maker’s point of view) also decreases the disparity. Why would we want to deviate from this policy to increase disparity *and* make the decision-maker worse off?

### Valuable but not crucial changes:

Rather than choosing two constraints without particularly compelling rationales, why not explore the full Pareto frontier of $E[g_{\sigma_A}(s,X)|S]$ for both groups? This would reveal whether either of the constrained solutions produced Pareto-dominated results (where both groups could be made better off). It would also allow the decision-maker to understand and navigate the trade-offs between good outcomes for each group.

Second, you can't really waive away the problem of causal inference. It's obviously a hard problem that won't be solved by this paper, but more discussion is needed to point out that understanding the conditional effect of actions on outcomes is very challenging in practice and might limit the applicability of this approach.

"These assumptions are not restrictive" is a strong claim. In practice, it is often very challenging to learn  the optimal policy from historical data because of unobserved confounders. That a certain partial ordering of nodes produces these conditional independencies does not help us understand whether the independences are restrictive because we're given no reason to believe the partial ordering is reasonable.

### Questions and minor changes:

“by focusing recruiting efforts in events organized by minority groups in relevant conferences and by choosing recruiting strategies that do not interact with group membership” aren’t these mutually exclusive? Targeted outreach efforts purposefully interact with group membership.

Is this a typo? “We created continuous actions corresponding to funding level decisions as A = <equation>, where E is the original percent of students taking the SAT/ACT exams (pre-college entry) in the dataset.” There is no E in the equation.

In Eq 1 the goal of the decomposition should be spelled out before the equation is presented. I was looking for where f, g, and h were defined earlier in the text.

The "regime indicator" is used prominently throughout the paper, but is only introduced via a citation. It should be described in the body of the paper so that readers unfamiliar with the concept can follow what is to come.

**Strengths And Weaknesses:**

The core ideas of this paper are good. This paper nudges the field closer to real-world applicability by focusing on whether actions have good outcomes and acknowledging that not all outcomes can be achieved with the actions available. The algorithmic fairness community would benefit greatly from taking these ideas on board.

However, significant changes to the presentation of the paper are needed to communicate these ideas convincingly. In particular, for a paper that intends to be pragmatically oriented towards the real world, the examples presented (both in the narrative and the experiments) are weak. ModBrk is introduced with a two-sentence example where a company is trying to recruit applicants for a job. This example is never returned to, the relationship between the space of actions and their impact on outcomes is not discussed, and the whole example only considers the company's goals, never what (if any) benefit applicants might receive.

The example introducing EqB is similarly weak. First of all, I don't think income tax systems are designed to optimize disposable income or savings levels. Then, the paper suggests that a new tax system might be required to produce the same benefit - relative to the status quo - for working moms as it does for other groups. It's not intuitive that this is desirable. For example, an increase in the US's Earned Income Tax Credit would fail this constraint, since it would disproportionately benefit working moms compared to those who don't work or aren't moms. In general, the constraints proposed by this paper (as opposed to the "pragmatic" considerations) are not well justified on normative grounds.

The examples in the experiments further confuse matters. In the IHDP experiments, the ModBrk constraint is defined to “break the moderation of the allocation of days in program by the group membership”. But the number of days in the program is the action, and ModBrk was originally defined with respect to the *outcome* (and, to me, that’s what makes it preferable over demographic parity).

A separate weakness is that the paper assumes away substantial challenges that would affect the method in practice. Most importantly, it is often extremely difficult to estimate the causal effects of actions or policies, and especially heterogeneity in those effects. The authors also acknowledge that their work is limited by the parametric assumptions needed to operationalize the EqB constraint and the simple decomposition used to enforce the ModBrk constraint.

---

### Review · Reviewer_wtQP · 2023-03-23

**Summary Of Contributions:**

This paper proposes two notions of outcome disparity control in the context of optimal policy learning, named ModBrk and EqB, and proposes two ways of estimating policies that satisfy these constraints. As is common in the fairness literature, the constraints are instantiated with a user-chosen epsilon "slack" parameter that trades off utility and fairness.

The paper includes extensive experiments that show that the proposed methods work in the sense that (1) they result in policies that (approximately) satisfy the targeted constraints, (2) raising the slack parameter in the constraints causes both the utility and the relevant disparity measure to rise, as expected, and (3) when the slack parameter is loose enough, their methods learn policies that improve over the baseline policy that generated the training data, as desired.

**Audience:**

Yes

**Claims And Evidence:**

No

**Requested Changes:**

I'd describe changes that address the first five points in the Weaknesses section above as critical. I think changes that address the next two points would strengthen the work. The last two points may be a matter of personal taste but will not change my evaluation.

**Strengths And Weaknesses:**

Strengths
---------
To the best of my knowledge, the two proposed outcome disparity measures are novel, which means that the accompanying methods and empirical results are substantially novel as well.

The outcome disparity measures have intuitive appeal, and I feel confident that they will hold interest for many readers.

The experiments are extensive and convincing, and they are conducted on an appropriate range of datasets.

The estimators and methods by and large make sense to me, and most aspects appear to be technically correct. As discussed below, there are several important points that are unclear to me, but I believe these can probably be addressed through greater clarity and without fundamental changes to the methods or results. For this reason, I am selecting "No" for now under the "Claims and Evidence" section, with the expectation that I will change it to "Yes" once these points are clarified.


Weaknesses
----------
Some of the assumptions and aspects of the methods are unclear to me. For this reason, I'm not entirely sure of the scope of the contribution, and I'm not 100% sure of the technical correctness in every instance. In general, I think many places in the paper could benefit from additional explanation. Specific points follow in roughly decreasing order of importance:

- "Assumption (a) enables us to learn the optimal policy based on historical data collected using a baseline policy..." This can't be true without an additional overlap assumption, right? There's no way to learn an optimal policy whose support in the action space is not contained in the support of the action space for the baseline policy, at least not without some other assumptions. The authors appear to acknowledge this in the "Action clipping" section, but there it's framed as a methodological concern rather than as an identifiability requirement. Either the optimal (fair) policy has to be assumed to satisfy this requirement, or the estimand has to be redefined to be the best fair policy within the set that does satisfy this requirement. Either way, this is potentially a pretty strong assumption/constraint, for example if the action space is continuous or the baseline policy was deterministic.

- I also don't quite understand how the action clipping is supposed to work. Isn't the proposed interval only defined for values of (s, x) that were observed at least twice in the training data? Does that mean this only works with features X that are discrete with a small number of levels relative to the size of the data?

- Is the baseline policy assumed to be known? That's implied by the expressions for the estimators, but it doesn't seem to be stated anywhere. I don't think the methods would have to change fundamentally if the baseline policy were unknown and had to be estimated from the data, but that would be important to address.

- I'm not previously familiar with the notion of a "regime indicator," here \sigma_A, and I'm a bit confused by the presentation. I understand it to be a parameter that defines the learned policy, as when it appears in the expression p(A | S, X; \sigma_A). But then assumption (a) puts it in an independence relationship, which suggests that it's a random variable. Since this quantity is crucial for everything that follows, this needs more explanation.

- It would be really helpful to provide some intuition about the decomposition in equation (1). Is the motivation here strictly methodological? As in, why bother thinking about nonzero versions of f(s, x) and h(a, x)? When these are both zero, then the ModBrk constraint reduces to (E[Y | S = s] - E[Y | S = s'])^2 \leq \epsilon, which looks a lot like demographic parity, though of course it applies to the outcome rather than the action. When f(s, x) and h(a, x) are nonzero, what does the constraint actually mean? Can we tell during the learning process what version of the constraint we're targeting on each iteration, or does it depend on how the neural net happens to be decomposing the left hand side of (1) into the right hand side at that moment. There's a mention in the conclusion of "a possible failure mode, where g simply subsumes f and h," which may be getting at this, but I don't actually know what that means. If this is a standard decomposition, then it would help to include some citations, but either way I think further explanation is needed.

- The top row of Figure 4 is very difficult to read.

- Equation (3) has "=" even though the requirement it instantiates says "approximately equal." Later on the usual slack parameter \epsilon is introduced. I was initially confused by this. I think it would be clearer to use \approx or just introduce \epsilon right away.

- To the best of my knowledge, the phrase "the fundamental problem of causal inference" was first used by Paul Holland (1986), and I think he deserves a citation. More substantially, I find it somewhat misleading to suggest that this problem applies to the EqB setting but not the ModBrk setting. Insofar as an optimal policy consists of an argmax over the set of available potential outcomes, only one of which is ever observed at a time, optimal policy learning always confronts this problem. The graph in Figure (a) assumes away the unmeasuring confounding aspect of this problem, but such a structural assumption is precisely a response to the fundamental problem. Otherwise we'd do optimal policy learning under much looser assumptions.

- I don't fully understand what the authors mean when they emphasize their pragmatic stance, e.g. "We take a pragmatic approach asking what we can do with an action space available to us and only with access to historical data." Isn't this the best that any policy learning approach can do? I know some of the papers in the Related Work section consider counterfactual worlds, but isn't that just a different route toward the same goal? I might've been missing the point.

---

### Decision · Action_Editors · 2023-07-27

**Recommendation:** Reject

**Comment:**

I do encourage the authors to resubmit a considerably revised version of the paper to TMLR.  I believe the paper will meet the criteria for acceptance once appropriately revised and re-reviewed for satisfaction of the “claims and evidence” criterion.

In revising the manuscript, I encourage the authors to address all of the points raised by reviewers that the authors in their replies said they would address.  In addition, I suggest a special focus on:

(i) Expanding on the motivating examples and experiments in a manner that provides sufficient detail to convince readers that the methods are in fact reasonably well motivated rather than entirely stylistic.  While the authors note they are wary of providing such detail as to transgress into the territory of “applying social science,” I don’t see a real risk of of that here.  Reviewer Z84d highlighted several issues with the examples as presented, which the authors have to some extent addressed in their replies.  I feel it is important to thoroughly address Z84d’s concerns because papers in the algorithmic fairness space are interesting to the extent that they address a compelling, well-motivated problem.  Were this paper being assessed on its merits as a contribution to, say, causal inference methodology or constrained optimization, it would fall well short of acceptance criteria for those domains.  So it is important here to provide sufficient detail to be convincing.  If upon striving to provide that detail the example starts to fall apart, one should then seek to find a different example.

(ii) Providing further normative justifications of the proposed criteria (ModBrk and EqB), both by elaborating on examples (see (i) above) and describing where such criteria might not reasonably apply.  I agree with Z84d that the justification/discussion provided in the current manuscript is insufficient.

(iii) Filling in missing (often implied) technical details.  For instance, there were questions about what’s observed and what’s latent, making explicit the no unmeasured confounding assumption, etc.  At the moment, the Conclusion and Method sections (e.g., Action clipping bit) do a lot of heavy lifting in clarifying the operating assumptions.  This makes the paper challenging to digest.  The assumptions should be clearly laid out earlier on, and can then be revisited in later sections.

(iv) Lastly,  I agree with Reviewer YCuy that it is essential to further clarify issues surrounding the under-specification of the ModBrk constraint.  Being more explicit about the “subtleties” of the decomposition (as discussed on the authors’ response) will go a long way toward resolving confusion.  Specifically, I reproduce here a comment from YCuy’s official recommendation:

However, the essential soundess issue remains—namely, the underspecification of the ModBrk constraint. The fundamental problem is this: because the decomposition $\mathbb{E}[Y \mid a, s, x] = f(s, x) + g(a, s, x) + h(a,s)$ is not identifiable, the constraint is vacuous.  By this, I mean that for any given policy $\sigma_0$, there exist $f_0$, $g_0$ and $h_0$ such that $\sigma_0$ satisfies the ModBrk constraint under this decomposition.  In particular, the decomposition, given fixed $\sigma_0$, is as given in my initial comment:

$$
f_0(s,x) = \mathbb{E}[Y_{\sigma_0} \mid s, x],  \qquad g_0(a,s,x) = \mathbb{E}[Y \mid a, s, x] - f_0(s, x), \qquad h_0(a, s) = 0
$$

Since $\sigma_0$ is fixed, $f_0(s, x)$ is a function of $s$ and $x$ alone.  [ModBrk is satisfied with $\epsilon = 0$ for $\sigma_0$ with this decomposition.]  This example appears to show that for any policy $\sigma_0$, one could adversarially decompose $\mathbb{E}[Y \mid a, s, x]$ such that the policy satisfies the ModBrk constraint with $\epsilon = 0$.

Clarifying why this is not an issue for the proposed approach will be important for the revision.

Note that some of these revisions will increase the length of the paper.  I think this is OK given that the paper is currently 9 pages long.  If it’s 11-12 pages long following revision, that is fine.

**Audience:**

**Would some individuals in TMLR's audience be interested in the findings of this paper?**

This paper considers a problem that the reviewers and I agree would be of interest to many in TMLR’s audience who are interested in questions of algorithmic fairness.  In particular, the work introduces two fairness/equity criteria and corresponding methods for reducing (or maintaining) disparity in outcomes while seeking to maximize overall outcomes through an improved policy.  This differs from prior recent work at the intersection of fairness and causality because the approach does not seek to model counterfactuals with race as the “intervention”, and thus in many ways relies on simpler assumptions.

All reviewers agreed that the submission meets this criterion.

**Claims And Evidence:**

**Are the claims made in the submission supported by accurate, convincing and clear evidence?**

This criterion entails assessing the “technical soundness as well as the clarity of the narrative and arguments presented.”  My primary reason for recommending a “Reject and resubmit”—that is, a reject decision with the invitation to resubmit a revised version as a fresh submission for a further round of review—is that the paper in its current form falls well short on clarity of narrative.   Many parts of the work are under-exposed, from the motivating examples, to the formal assumptions, to descriptions of how the methods relate to other work in the literature.  It’s not a paper that’s easy for even expert readers to readily understand from a thorough reading.  And that’s not because the technical concepts presented are particularly complex; it’s a reflection of the lack of clarity and poor exposition.  Many of the core issues have been pointed out through the reviewers’ extensive comments.  I believe the issues are too pervasive to be triaged through a minor revision process.

Two of the three reviewers in their final recommendation indicated they believe the submission continues to fall short on this criterion.  I agree.

**Resubmission Of Major Revision:**

The authors may consider submitting a major revision at a later time.